# Deletion of the Serotonin Receptor 7 Gene Changed the Development and Behavior of the Mosquito, *Aedes aegypti*

**DOI:** 10.3390/insects13080671

**Published:** 2022-07-25

**Authors:** Miaozhen Li, Lei Zhang, Yuchen Wu, Yixun Li, Xin Chen, Jing Chen, Qiuhui Wang, Chenghong Liao, Qian Han

**Affiliations:** 1Laboratory of Tropical Veterinary Medicine and Vector Biology, School of Life Sciences, Hainan University, Haikou 570228, China; 18071010210009@hainanu.edu.cn (M.L.); zhanglei@hainanu.edu.cn (L.Z.); 20190881310166@hainanu.edu.cn (Y.W.); lyx19271927@163.com (Y.L.); 20197201310035@hainanu.edu.cn (X.C.); 19071010110002@hainanu.edu.cn (J.C.); 19071010210011@hainanu.edu.cn (Q.W.); 2One Health Institute, Hainan University, Haikou 570228, China

**Keywords:** 5-HT, 5-HT receptors, *Aedes aegypti*, CRISPR-Cas9, behavior

## Abstract

**Simple Summary:**

*Aedes aegypti* can transmit various diseases and is a serious public health risk to humans. Due to the abuse of insecticides, pesticide resistance has become a new challenge in mosquito management. The 5-HT receptor is a potential target for novel insecticides. However, in *Ae. aegypti*, the function of the 5-HT receptor is not fully understood. There are three types of 5-HT receptors in insects, namely 5-HT_1_, 5-HT_2_, and 5-HT_7_. In this study, we used real-time PCR to find that male mosquitoes express higher levels of the 5-HT_7A_ receptor than females and all larvae, and that expression was higher in the appendages but not in the head. We obtained 5-HT_7A_ mutants using CRISPR-Cas9 technology. The mutant larvae showed changes in growth and development such as body contraction, reduced motility, and reduced stress response to external stimuli.

**Abstract:**

Serotonin (5-HT) plays a vital role in many physiological processes in insects, regulating physiological activities such as growth and movement through multiple 5-HT receptors (5-HTRs), which were potential targets for some new insecticides. However, the specific function of individual 5-HTRs in *Ae. aegypti* is still unclear. In this study, we investigated the function of the 5-HT_7A_ receptor during *Ae. aegypti* development. 5-HTR_7A_ transcripts were detected at all stages of development by real-time PCR. The results indicated that the gene expression was highest in the limbs (*p* < 0.01). We also generated 5-HTR_7A_ mutant mosquitoes using CRISPR-mediated gene editing. The mutants had an abnormal phenotype at the larval stage, including an aberrant head-to-chest ratio and decreased motor activity. The mutant pupae developed abnormally, and most died (56.67%) (*p* < 0.0001). Using external stimuli to larvae and pupae with abnormal phenotypes, we found the mutant G1 and G2 generations responded to external stimuli in a longer time than the wild-type (WT) mosquitoes, and most of the mutants were 2 to 3 s slower than the WTs to respond to external stimuli (*p* < 0.01). Due to higher mortality, mutant larvae and pupae had fewer numbers than the WTs. The egg hatching rate of mutant G1 and G2 generations was lower than that of the WTs (*p* < 0.01). The expression level of 5-HTR_7A_ in the mutants decreased by about 65% compared with the control group using real-time PCR (*p* < 0.05). In all, the 5-HT_7A_ receptor plays an important role in the metamorphosis, development and motor function of *Aedes aegypti*.

## 1. Introduction

Serotonin (5-hydroxytryptamine, 5-HT) is a critical neurotransmitter widely distributed in the central and peripheral nervous systems of vertebrates and invertebrates [1]. 5-HT regulates many essential physiological and behavioral processes such as locomotory rate, reproduction, sex activity, emotion, sleep, and appetite by binding to 5-HT receptors (5-HTR) [2,3,4,5,6,7]. 5-HTR typing is complex. At least 14 subtypes in 7 major groups have been identified in the human genome. Except for 5-HTR_3_, which is a ligand-gated ion channel, all other 5-HTRs are G-protein-coupled receptors (GPCRs), according to their conserved amino acid sequence and second messenger activation system [5]. 5-HT binds to the extracellular domain of the receptor, and the G protein-binding domain is intracellular. 5-HT binding induces conformation changes in the G protein-binding domain, which activates G proteins, and downstream adenylate cyclase (AC) and phospholipase C (PLC) to mediate cellular responses [8].

5-HT and 5-HTRs play key regulatory roles in the immune system of insects. 5-HTR_1B_ and 5-HTR_2B_ can mediate phagocytosis in *Pieris rapae*. This study also demonstrated that insects regulate nervous and immune system interactions through 5-HT at the molecular level. Thus, it may be possible to use 5-HT-targeting drugs to specifically interfere with the insect immune system and enhance insecticide effectiveness [9]. Studies have also shown that the neuronal activation and subsequent release of large amounts of 5-HT inhibit feeding and mating behavior in *Drosophila melanogaster* [4]. Similarly, injection of 5-HT into the brain of bees can inhibit feeding behavior, whereas injection into the intestine can stimulate intestinal muscles [10]. 5-HTR_2B_ knockout using CRISPR-Cas9-mediated gene editing resulted in growth inhibition in *Ae. aegypti* and reduced lipid accumulation in adult mosquitoes [11]. At present, the design, synthesis, and bioassay analysis of 5-HTR_1A_ agonist derivatives have also shown that 5-HTR_1A_ can affect insect growth and larval activity [12]. These results confirmed that 5-HTR could be a drug target in harmful insects. Due to worldwide pesticide abuse, resistance to traditional small-molecule insecticides has become a challenge for controlling arthropod-borne infectious diseases. Therefore, insecticide chemicals with new modes of action and activity against pest populations are required as alternatives. Small-molecule insecticides that destroy the targets of arthropod GPCRs have gradually become the focus of attention and may be used to develop next-generation insecticides [13,14].

The *Ae. aegypti* mosquito is a severe threat to public health as it is responsible for spreading yellow fever, dengue, and Zika viruses [15]. 5-HT plays a crucial role in various physiological processes in insects through binding to 5-HTRs [1]. Thus, 5-HTR could be a potential target for new insecticides. However, the function of 5-HTR in *Ae. aegypti* has not been fully elucidated. In previous studies, we constructed the spatiotemporal expression profiles of six 5-HT receptors in *Aedes aegypti* and found that the expression trends of these 5-HT receptors did not differ during growth and development. However, in the spatial expression profile, only the 5-HT_7A_ receptor was the highest expressed one in the appendages and expressed little in the head, so we speculated that the 5-HT_7A_ receptor might be related to the motility or host recognition ability of mosquitoes. Therefore, to investigate its function in *Ae. aegypti*, CRISPR-Cas9-mediated gene editing was used to introduce a deletion in the 5-HTR_7A_ gene. We found that the mutants had an abnormal phenotype at the larval stage, the pupae could not develop normally, and most died. Due to the higher mortality of mutant larvae and pupae, the number of later generations was reduced compared with WT mosquitoes. We also found that the egg hatching rates of mutant G1 and G2 generations were lower than those of WT mosquitoes.

## 2. Materials and Methods

### 2.1. Insects

The WT and mutant *Ae. aegypti* Rockefeller strain mosquitoes were kept in the insectary at Hainan University [16]. The egg paper was placed in a feeding basin with added water and fodder. Larvae were reared in water with fish food, and the adults were kept at 26 °C, with relative humidity of 70% under a 16 h (light):8 h (dark) photoperiod, and were provided with an 8% sugar source [14].

### 2.2. Phylogenetic Analysis

Clustal omega online software was used to analyze the alignment of amino acid sequences and to construct the evolutionary tree by using the neighbor-joining method. Bootstrap was set to 1000, and the other parameters used were set as default values.

### 2.3. qPCR

To analyze the temporal and spatial expression profiles of *Ae. aegypti* 5-HTR_7A_ (GenBank accession number: AAEL025125) during development, we collected three days old adult mosquito samples from the head, thorax, abdomen, legs, and different development stages (white pupae: newly formed pupae before turning to black; black pupae: pupae with black color after an hour). Total RNA was isolated for three independent replicates, and complementary DNA (cDNA) was synthesized using the TAKARA reverse transcription kit. qPCR was performed as described previously [16] using a LightCycler 480 system (Roche Applied Science, Mannheim, Germany) and SYBR green Master I (Roche) according to the manufacturer’s instructions, with the following cycling conditions: initial denaturation at 95 °C for 30 s followed by 40 cycles of 95 °C for 5 s, and 60 °C for 30 s. At the end of the PCR reaction, a melting curve was generated to assess the possibility of undesirable side products. The threshold cycle number (CT) was determined and was used for comparative quantitative analysis. The primer pair that amplifies a fragment of the relatively abundant and constitutively expressed *Ae. aegypti* ribosomal protein S17 (*RPS17*) gene (GenBank accession number: AAEL025999) was used to normalize the results of variable target genes and to correct for sample-to-sample variations, which was a relatively abundant and constitutively expressed gene. The primer sequences are shown in Appendix A.

### 2.4. CRISPR-Cas9-Mediated Knockout of the Ae. aegypti 5-HTR_7A_

The single-guide RNA (sgRNA) designed for the 5-HTR_7A_ gene was based on the principle of N20NGG using an online tool (http://chopchop.cbu.uib.no/) (accessed on 20 July 2019). A single exon of the *Ae. aegypti* 5-HTR_7A_ was selected as a target. A template for 5-HTR_7A_ was generated by template-free PCR using DNA polymerase. The PCR products were resolved by agarose gel electrophoresis, and bands of about 100 bp could be identified. Then, PCR products were purified with the Gene Jet PCR Purification Kit (Sangon Biotech, Shanghai, China). In vitro transcription of sgRNA was performed using the Megascript T7 Kit (Thremo Scientific, Waltham, MA, USA) following the manufacturer’s protocol and incubating for 12 h at 37 °C. sgRNA was purified using the MegaClear column purification kit (Thremo Scientific, USA) following the manufacturer’s protocol [16]. To remove any residual salt/ethanol/other impurities from the sgRNA before microinjection, a final purification was performed as described previously [17].

To generate stable germline mutations, we injected sgRNA (200 ng/μL) and Cas9 protein (300 ng/μL) into the posterior end of the embryos. A total of 180 embryos were injected. The embryos hatched 7 d after injection. A total of 14 embryos hatched, and normally emerged adult mosquitoes were mated with WT mosquitoes. To generate G1 progeny, G0 females were blood-fed and placed into the oviposition tubes individually. Genomic DNA was extracted using the Animal Genomic DNA Purification Kit (Sangon, Shanghai, China). The target fragment was amplified by PCR and then purified using Column PCR Product Purification Kit (Sangon, Shanghai, China). Finally, the mutants were screened using Sanger sequencing (Sangon, Shanghai, China). The primer sequences of sgRNAs are shown in Appendix A.

### 2.5. Evaluation of Stress Response the Mutants

The mutant and WT mosquitoes were stimulated with a Pasteur pipette, and the whole reaction process was recorded with a camera. The stress response of mosquitoes to external stimuli was calculated based on their recorded reaction time.

### 2.6. Statistical Analysis

All data were analyzed with GraphPad Prism Software version 7. For qPCR analysis, 2-ΔΔCt was used to calculate the differences in expression of multiple genes [18]. T-tests and two-way analysis of variance processed were used for statistical analysis (* *p* < 0.05, ** *p* < 0.01, *** *p <* 0.001, **** *p <* 0.0001).

## 3. Results

### 3.1. Phylogenetic Analysis of Ae. aegypti 5-HTR Proteins

According to the study, insect 5-HT receptors can be classified into 5-HT_1_, 5-HT_2_, and 5-HT_7_ type GPCRs based on the conserved amino acid sequence and the activated second messenger system [1]. Therefore, we performed evolutionary analysis of *Ae. aegypti*. The phylogenetic tree of six 5-HTR proteins was constructed using the adjacency method of MEGA 7 (Figure 1). *Melanaphis sacchari*, a Hemiptera insect [19], was used as an outgroup. 5-HTR_2A_ and 5-HTR_2B_ of *Drosophila melanogaster* were converted to primitive 5-HTR. 5-HTRs of *Ae. aegypti* evolved into three subfamilies. *Ae. aegypti* and *Anopheles albimanus* 5-HTR_1D_ were in the same cluster. *Ae. aegypti* 5-HTR_1A_ formed a sister group with *Drosophila melanogaster* 5-HTR_1A_. *Ae. aegypti* 5-HTR_7A_ and 5-HTR_7B_ were clustered with *Drosophila melanogaster* 5-HTR_7_. 5-HTR_2B_ and 5-HTR_1D_ of *Aedes aegypti* formed a sister group.

### 3.2. Temporal and Spatial Expression Profiles of 5-HTR_7A_ Genes

5-HTR_7A_ temporal and spatial expression profiles were examined using qPCR. The transcripts throughout the mosquito lifecycle of 10 different development stages were investigated, with high 5-HTR_7A_ expression detected in the male adults and male black pupae. 5-HTR_7A_ gene expression increased gradually during growth and development. In adults, the transcript was more abundant in the legs than in other tissues. Interestingly, 5-HTR_7A_ gene expression was not detected in the head (Figure 2).

### 3.3. CRISPR-Cas9-Mediated 5-HTR_7A_ Gene Knockout

*Ae. aegypti* 5-HTR_7A_ is a typical GPCR with seven transmembrane domains. The 5-HTR_7A_ binding site for extracellular 5-HT is between the 5th and 6th transmembrane domains [20]. Therefore, the 5th transmembrane region was selected as the deletion site (Figure 3a). The mixture of purified sgRNA and Cas9 protein was injected into the fresh *Ae. aegypti* eggs. In the G0 generation, female mosquitoes were mated with WT male mosquitoes. In the G2 generation, the offspring of the mutated G1 generation were selected for random pairwise mating. The mutants were screened by Sanger sequencing. One hundred and eighty *Ae. aegypti* eggs were injected with the mixture of sgRNA and Cas9 protein. About 7.7% (14/180) of eggs were hatched, and 71.43% (10/14) developed into the adult stage. 5-HTR_7A_ gene deletions, as confirmed through Sanger sequencing analysis, occurred in about 60% (6/10).

All mutations were detected using a sequencing peak map, and peaks detected in G0 and G1 generations showed impurities (Figure 3b). We compared the 5-HTR_7A_ gene sequences of mutant and WT mosquitoes using the online software DSDecodeM (DSDecodeM (scau.edu.cn) accessed on 12 January 2020) and found that deletion of two bases occurred at the target position in the mutants (Figure 3c) [21]. The protein sequence of the mutant 5-HTR_7A_ was completely changed after amino acid 291 compared with the WT (Figure 3d). TMHMM online software (TMHMM—2.0—Services—DTU Health Tech) was used to predict whether changes were found in the transmembrane domain of the mutant 5-HTR_7A_ mosquitoes. The results showed that the mutant lacked three transmembrane regions (Appendix A), which resulted in the deletion of the extracellular 5-HT binding site of 5-HTR_7A_, and the inactivation of the intracellular G protein binding site.

### 3.4. Morphological Changes in 5-HTR_7A_ Mutant Mosquitoes

Compared with the WTs, the mutants had an abnormal head-to-chest ratio in the 4th instar larvae. In the WTs, the head-to-chest ratio was about 1:2, but in the mutants, we observed that the head-to-chest ratio was about 1:1 (Figure 4a). In addition, the larval body contracted and twitched. We also found that the larval activity was weakened, and larvae eventually lost the ability to sink to the bottom and died. When molting from the 4th instar larval stage into pupae, an abnormal phenotype in pupae was observed (Figure 4b), which led to the death during the pupal stage. We analyzed the number of larvae and pupae. The 5-HTR_7A_ mutants showed a significantly higher proportion (56.67%) of abnormally developed larvae and pupae by comparing the numbers in the WTs (Figure 4c).

### 3.5. Stress Response of 5-HTR_7A_ Mutant Mosquitoes

There were obvious differences in the motor activity of 5-HTR_7A_ mutants at the larval and pupal stages compared with the WTs. Mutant larvae and pupae had a slower reaction time following an external stimulus (Figure 5). Most of the mutants were 2 to 3 s slower than the WTs to respond to external stimuli.

### 3.6. Effects of 5-HTR_7A_ Mutations on Egg Hatching in Offspring

As shown in Figure 6, the egg hatchability of offspring produced by adult mutant mosquitoes was affected. The egg hatching rate of the mutants was 50% lower than that of the WTs.

### 3.7. 5-HTR_7A_ Expression in Mutant Mosquitoes

5-HTR_7A_ expression in the larvae with obvious phenotypes in the mutant group was analyzed by qPCR. The results showed that the expression of the mutant 5-HTR_7A_ was reduced by 65% compared with the WTs (Figure 7).

## 4. Discussion

5-HT is a neurotransmitter widely found in animals. It plays a role in regulating emotions, sleep, reproduction, feeding, movement, fighting, and other behaviors. Compared with a single 5-HTR_7_ in *Drosophila melanogaster*, we found that *Ae. aegypti* appears to have two 5-HTR_7_ proteins [22,23]. Therefore, we performed sequence alignment and homology analysis between AAEL025125 (5-HTR_7A_) and AAEL027242 (5-HTR_7B_). The results showed that the identity and similarity of the two were 61.44% and 86.23%, respectively (Appendix A). Phylogenetic analysis also indicated that Aae5-HTR_7A_, Aae5-HTR_7B_, and Dro5-HTR_7_ were clustered into one group (Figure 1). Secondary structure prediction demonstrated that α-helix and β-turn contents were almost identical for both proteins (Appendix A). The temporal and spatial expression profile of 5-HTR_7A_ showed that it was expressed during all growth stages (Figure 2). The spatial expression profile of 5-HTR_7B_ was similar (Appendix A), but its overall expression level was lower than that of 5-HTR_7A_. We speculated that these two receptors could play an important role throughout the lifecycle of *Ae. aegypti*. In this study, we found that 5-HTR_7A_ was not expressed in the head (Figure 2b). Compared with 5-HTR_7B_, 5-HTR_7A_ expression was higher in the limbs. The differences in spatial expression profiles reflect the seemingly complementary relationship between the two receptors. Studies found that serotoninergic neurons exist in the maxillary palps of female mosquitoes [24]. Sharon et al. showed that 5-HTR_7B_ expression remained unchanged pre-and post- blood meal [25]. The function of 5-HTR_7_ in *Ae. aegypti* has not been investigated up to now.

In this study, we found that 5-HTR_7A_ was not expressed in the head, and it was the only 5-HTR with high expression in the limbs. Therefore, it can be speculated that this receptor may be involved in the perception of external temperature and humidity or in detecting scent [26]. To explore the function of 5-HTR_7A_, we used CRISPR-Cas9-mediated gene editing to introduce a deletion in the *5-HTR_7A_* gene. The mutant heterozygote mosquitoes showed a high proportion of larval dysplasia. The specific phenotype was that larvae contracted and twitched, and their ability to respond to external stimuli was reduced. Larvae and pupae with these phenotypes do not normally develop into an adult. The phenotype at the larval stage also appeared at the pupal stage. The egg hatching rate of the mutant offspring was significantly different than that of the WTs. The severe phenotypes that led to abnormal phenotypes consisted of a large proportion of the mutants, about 77.3%. The mild phenotype that caused a low hatching rate of eggs made up only 22.7% of the mutants. 5-HTR_7A_ mutation decreased receptor expression by 56%, and the binding efficiency of extracellular 5-HT to the receptor may be reduced accordingly. This would inhibit the interaction between 5-HTR_7A_ and the G protein αs subunit leading to reduced AC activation and decreased production of the second messenger cAMP [8].

Research on 5-HTR in insects is also focused on the development of efficient insecticides. Currently, chemical pesticides are the main means of pest control, but their use has created issues such as pesticide resistance. Therefore, it is necessary to develop new insecticides. This study showed that 5-HTR_7A_ plays an important role in the metamorphosis of *Ae. aegypti* and may become a new target for insecticides. The pharmacological properties of 5-HTRs in insects are different from those in mammals. 5-HTRs also have different pharmacological properties in different insects. Studies have found that 8-hydroxy-2- (di-n-propylamino) tetralin (8-OH-DPAT) can activate 5-HTR_1A_ and 5-HTR_7_ in mammals [27,28,29,30]. In some insects, 8-OH-DPAT had an excitatory effect on 5-HTR_1A_, 5-HTR_1B_, 5-HTR_2A_, 5-HTR_2B_, and 5-HTR_7_ [31,32,33,34,35,36]. The binding affinity of homologous receptors to 5-HT in different insects also varies. One study found that the EC_50_ of 5-HTR_1A_ in *Apis mellifera* was only 16.9 nM, but the EC_50_ of 5-HTR_1_ in *Periplaneta americana* was 130 nM [37]. Various 5-HTRs can play multiple physiological functions after being activated by 5-HT. At present, there are still many gaps in the functional studies of insect 5-HTRs, and future studies on insect 5-HTRs and their functions will provide new insights into their functions, mechanism and applications in new pesticide discovery. Based on the differences in the pharmacological properties of 5-HTRs in different insects, identifying drugs with high selectivity to 5-HTRs in pests and harmless organisms will be another research focus.

## 5. Conclusions

Mosquito 5-HTR_7A_ was highly expressed in male mosquitoes compared to all instar larvae and female mosquitoes, with high expression in the appendages but not in the head. The mutant larvae of 5-HTR_7A_ using CRISPR-Cas9 techniques showed physical contractions, reduced motility, and reduced stress response to external stimulation during their growth and development. In addition, larvae with these phenotypes could not develop into adult mosquitoes, and the egg hatching rate of the mutant offspring was significantly reduced compared with WTs.

## Figures and Tables

**Figure 1 insects-13-00671-f001:**
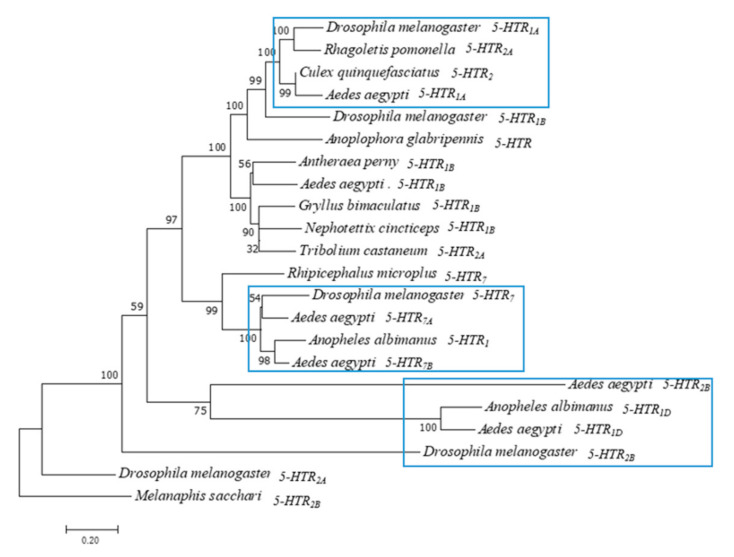
Phylogenetic tree of *Ae. aegypti* 5-HTR proteins. Phylogenetic tree, branch length and bootstrap values of serotonin receptors of *Drosophila melanogaster*, *Ae. Aegypti* and other insects. The numbers above the branches represent the bootstrap values for each branch (1000 replications, significantly support for the related sequences common to a node). Branch lengths are scaled. (Scale bar indicates 0.2 units of branch length).

**Figure 2 insects-13-00671-f002:**
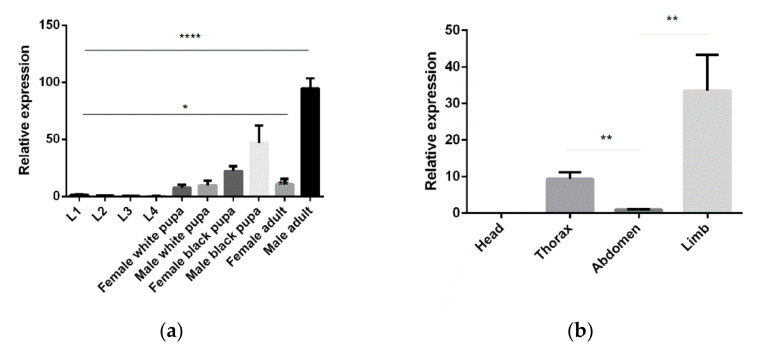
*Ae. aegypti* 5-HTR_7A_ temporal and spatial expression profiles. (**a**) 5-HTR_7A_ mRNA levels in 10 mosquito developmental stages, including eggs, larvae, and adults. (**b**) The differential expression levels of 5-HTR_7A_ were analyzed in the head, thorax, abdomen, and limbs. Statistical analysis of the differential expression profiles was performed using two-way analysis of variance processed by GraphPad software (* *p* < 0.05, ** *p* < 0.01,**** *p* < 0.0001; mean ± *SD*; n = 3).

**Figure 3 insects-13-00671-f003:**
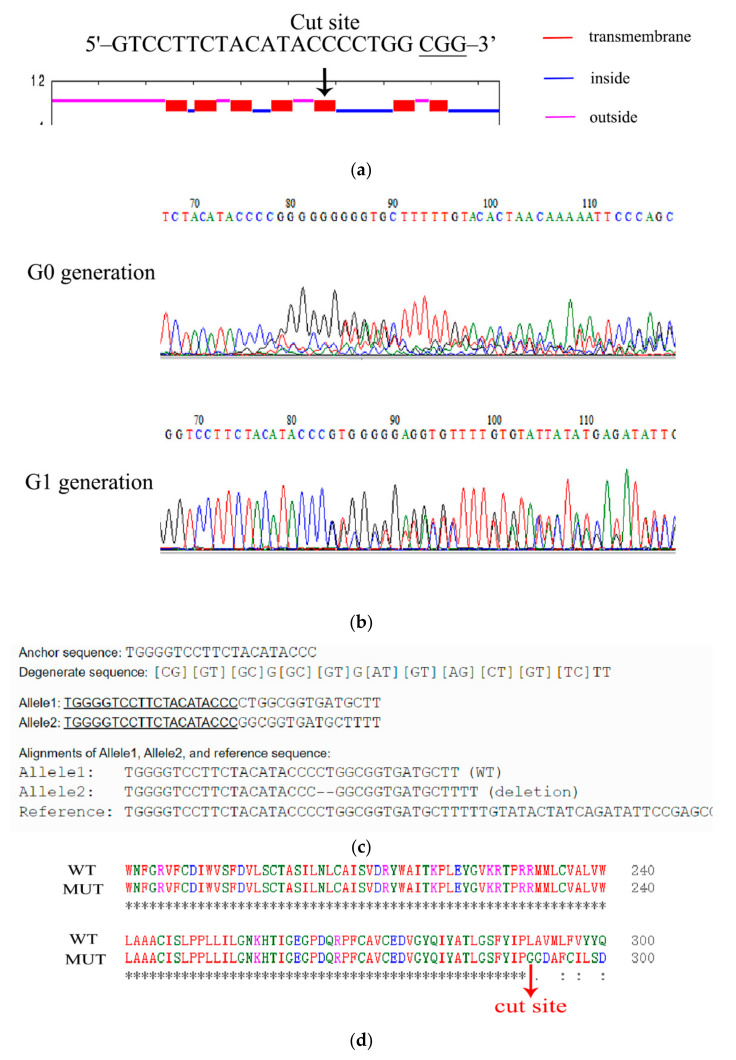
Generation of *Ae. aegypti* 5-HTR_7A_ mutants. (**a**) Location of CRISPR-Cas9-mediated gene deletion target. Transmembrane, intracellular, and extracellular domains are in red, blue, and purple, respectively. (**b**) Sequencing results for 5-HTR_7A_ mutants showed that the G0 generation was a chimera and G1 was a heterozygote. (**c**) Sequences from WT and mutant mosquitoes were compared using the online software DSDecodeM. Alleles 1 and 2 represent a normal and a mutated chromosome, respectively. (**d**) Protein sequence alignment between WT and mutant mosquitoes.

**Figure 4 insects-13-00671-f004:**
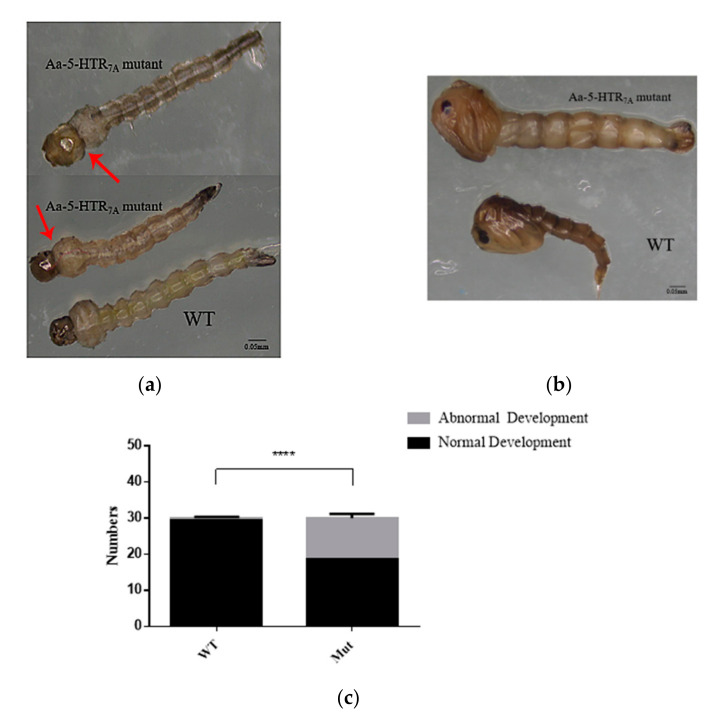
Phenotype of *Ae. aegypti 5-HTR_7A_* mutants at different developmental stages. (**a**) Phenotypic changes of the mutant larvae. The head-to-chest ratio was 1:2 in WT larvae and 1:1 in mutant larvae. The arrows refered to the position of the head-to-chest of WT larvae and mutant larvae respectively. (**b**) Phenotypic differences between the WTs and the mutants at the pupal stage. (**c**)The gray color in the bar shows numbers of abnormally developing *Ae. aegypti* larvae and pupae. The black color shows numbers of normally developing *Ae. aegypti* larvae and pupae. Data were plotted as mean ± SEM (n = 30 larvae and pupae per strain). Stars indicate statistical significance analyzed using a T-test (**** *p* < 0.0001).

**Figure 5 insects-13-00671-f005:**
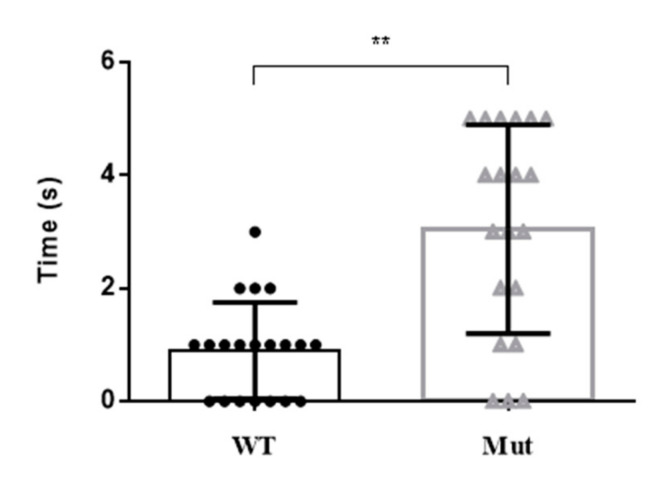
Motor activity changes in *Ae. aegypti 5-HTR_7A_* mutants after external stimulation. The black circles and the gray triangles represent the WTs and mutants, respectively. Data were plotted as mean ± *SEM* (n = 20 larvae and pupae per strain). T-tests were performed using Graph-Pad software (** *p* < 0.01).

**Figure 6 insects-13-00671-f006:**
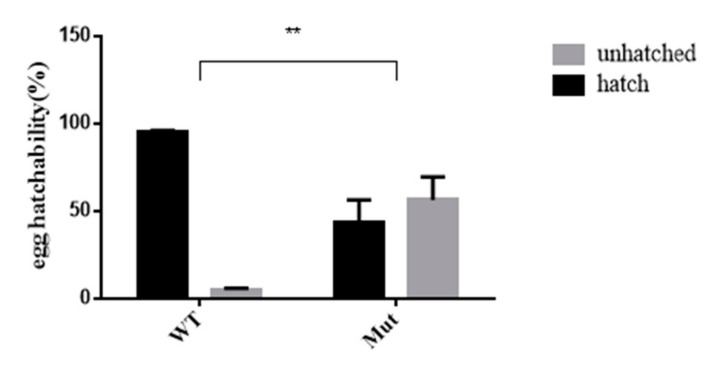
Comparison of hatchability between *Ae. aegypti* 5-HTR_7A_ mutants and WT. The gray and black bars indicate percentages of unhatched and hatched eggs, respectively. Stars indicate statistical significance analyzed using a T-test (** *p* < 0.01).

**Figure 7 insects-13-00671-f007:**
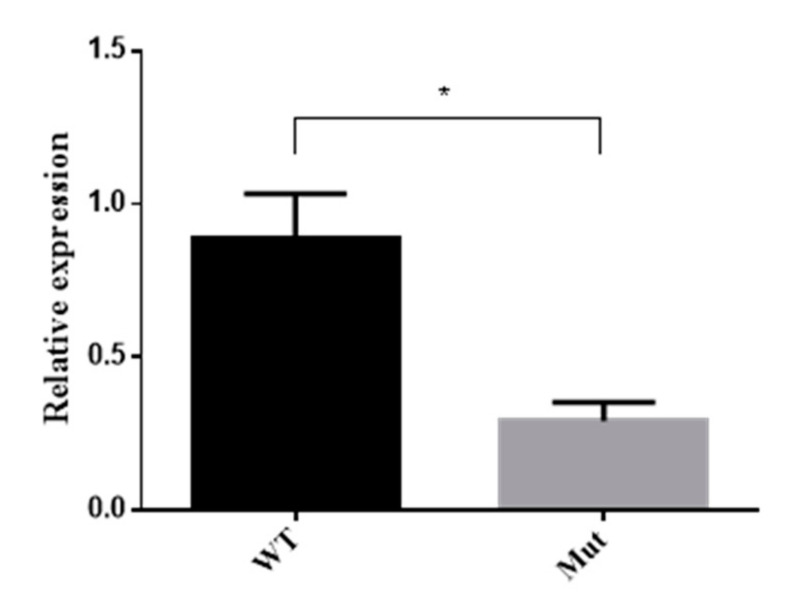
5-HTR_7A_ expression of mutants by qPCR. WT was used as the control group, and *RPS17* was used as a housekeeping gene. Black bars indicate WTs and grey bars indicate mutants. A T-test was performed using Graph-Pad software (* *p* < 0.05).

## Data Availability

The data used to support the findings of this study are included in the article.

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
