# Peer review of "Deletion of the Serotonin Receptor 7 Gene Changed the Development and Behavior of the Mosquito, Aedes aegypti"

_insects, 2022, doi:10.3390/insects13080671_

Round 1

Reviewer 1 Report

The paper (Insects 1793699) analyzes transcription of 5-HT7A (serotonin) in Aedes aegypti, then knocks out 5-HT7A gene using Crispr-cas9 gene editing. 5-HT7A was expressed mainly in mosquito limbs and males. Mosquito knockout mutants were developmentally challenged suggesting 5-HT7A receptors may be targets for new insecticides. The work describes a typical approach for modern gene function studies. Unclear why 5-HT7A was chosen among the multiple Aedes aegypti 5-HT receptors. There are a number of typos that will be listed below. The abstract as presented does not give statistical outcomes.  Stating that "expression was highest in the limbs" or later stating "....higher mortality,  in mutant larvae and pupae had fewer numbers" is somewhat vague and stating p values would give the reader a better sense of results. 

Typos.....

1) Line 90...'melt curve', change to 'melting curve'.

2) Line 133....'basing' change to 'based'.

3) Line 136......italicize Melanaophis saccharin

4) Line 136.....Is  M. saccharin really closely related to Aedes aegypti?

5) Line 137....italicize Drosophila melanogaster

6) Line 141.........delete "into a cluster"

7) Line 146 (Fig. 1 legend).....change "boostrap" to 'bootstrap"

8) Line 154-expression is undetectable in the head. How do the authors know the outcome is not due to eye pigmentations (shown by others to sometimes cause false negatives by inhibiting qPCR).  

9) Line 159 (figure 2 legend)....change "tow-way" to "two-way"

10) Line 166.....drop final "s" in offsprings

11) Line 273 Suggested rewrite...."Mosquitoes with these phenotypes did not develop into normal adults'.

12) Unclear as to specific phenotypes of male or female mutants, given that 'highest' expression of 5-HT7A was in male limbs.

13) Given that 5-HT's are widely found in animals, how could one develop an insecticide targeting only 'bad' insects. Future studies?

Reviewer 2 Report

Comments to authors

The study by Li et al. performed developmental expression profiling of one of the serotonin receptors (5-HTR7A) and edited using CRISPR in the disease vector, Aedes aegypti. The authors notice knockout of this gene affected the development of larva and pupa and egg hatching rate. The data appear to be sound, however, I noticed some issues with the data which are listed below.

Major comment:

Fig. 7. The authors show that the 5-HTR7A mRNA levels were reduced in the mutants. It is surprising how could mutating a coding region of a target by knockout affects its mRNA expression?  

 Line No. 270-271: The 5-HTR7A basal expression levels were very low during larval stages compared to its expression levels in pupal and adult stages. But authors observe high larval dysplasia. I wonder these effects are because of some non-target knockout effects.

Line No. 273: The authors mentioned that larva or pupa with abnormal phenotype could not develop into adults. However, they observed adults emerging from knockout mutants. I assume these adults are probably in the heterozygous state. It is interesting to check is there any difference in the mutation rate between the severe phenotype and milder phenotype showing mutant insects. This helps in understanding whether 5-HTR7A plays an indispensable role in mosquito development or not. 

Minor comments: 

Line No: 23: Include a concluding sentence at the end of the abstract to summarize the study outcome.  

Line No. 150: It is necessary to describe why only the 5-HTR7A gene was chosen among several isoforms of HTRs for expression studies and knockout. 

Line No. 150-152: How many hours old pupa is considered white and black? This information must be included. 

Line No. 159: I found several spelling errors like “tow-way analysis…” I advise authors to proofread and correct misspelled words. 

Fig. 2. The authors need to provide from which stage the head, thorax, abdomen, and limbs were dissected for RNA extraction. 

Fig. 4. In which stage of larva/papa, do the abnormal phenotypes start to appear? What happened to abnormally developed larvae/pupae? Are they survived?

Fig. 6. Knockout of 5-HTR7A has any effects on egg-laying capacity? The authors comment on this. 

Round 2

Reviewer 1 Report

Author's responded satisfactorily to all questions/edits. This paper should be accepted for publication

Two edits:

1) Line 294: ....change to "do not normally develop into adults."

2) Line 298 change to "consisted of a large ....".

Author Response

Response to Reviewer 1 Comments:

Thank you for your advice. 

1) Line 294: ....change to "do not normally develop into adults."

Response: We corrected it (P9, L294).

2) Line 298 change to "consisted of a large ....".

Response: We corrected it (P9, L298).

Reviewer 2 Report

The authors have sufficiently addressed the concerns raised in the review of the initial submission. 

Author Response

Thank you for your comments.

This manuscript is a resubmission of an earlier submission. The following is a list of the peer review reports and author responses from that submission.